# Rapid and Simultaneous Authentication of Six Laver Species Using Capillary Electrophoresis-Based Multiplex PCR

**DOI:** 10.3390/foods13030363

**Published:** 2024-01-23

**Authors:** Seung-Min Yang, Jun-Su Kim, Eiseul Kim, Hae-Yeong Kim

**Affiliations:** Institute of Life Sciences & Resources, Department of Food Science and Biotechnology, Kyung Hee University, Yongin 17104, Republic of Korea; ysm9284@gmail.com (S.-M.Y.); jwater0415@gmail.com (J.-S.K.); eskim89@khu.ac.kr (E.K.)

**Keywords:** laver, seaweed, multiplex PCR, capillary electrophoresis, simultaneous detection, food authentication

## Abstract

Lavers are typically consumed in dried or seasoned forms. However, commercially processed lavers can lead to seafood fraud because it is impossible to authenticate the original species based on morphological characteristics alone. In this study, we developed a capillary electrophoresis-based multiplex polymerase chain reaction (PCR) to authenticate six different laver species. The species-specific primer sets to target the chloroplast *rbcL* or *rbcS* genes were newly designed. We successfully established both singleplex and multiplex conditions, which resulted in specific amplicons for each species (*N. dentata*, 274 bp; *N. yezoensis*, 211 bp; *N. seriata*, 195 bp; *N. tenera*, 169 bp; *N. haitanensis*, 127 bp; *P. suborbiculata*, 117 bp). Moreover, the assays were sensitive enough to detect DNA ranging from 10 to 0.1 pg of DNA. The optimized capillary electrophoresis-based multiplex PCR was successfully applied to 40 commercial laver products. In addition to detecting the laver species as stated on the commercial label, the assay discovered cases where less expensive species were mixed in. With its advantageous properties, such as short amplicon size, high specificity, and superior sensitivity, this assay could be used for the authentication of the six laver species.

## 1. Introduction

Lavers have been cultivated for a long time in Asian countries, particularly in Korea, Japan, and China. The increasing awareness of their health benefits and the globalization of processed food products have significantly increased worldwide consumption [1]. Lavers are traditionally known as kim in Korea, nori in Japan, and zirai in China. They are generally consumed as processed foods, such as kimbab (seasoned laver with rice roll), zirai tang (laver soup), yakitori (roasted laver), and gim snack (seasoned laver) [2].

From a nutritional point of view, lavers are distinguished by their high fiber and mineral content, low fat levels, and relatively high protein concentrations. Lavers contain porphyran and vitamin B12, which are not found in other seaweeds. Porphyran, a unique and distinctive fiber, has bioactivities such as antioxidant and anticancer effects [3,4]. Additionally, lavers produce exceptional amounts of vitamin B12, making them an excellent option for addressing vitamin B12 deficiencies, such as methylmalonic acidemias, in vegan diets through the consumption of laver [5].

From a taxonomic perspective, laver is a type of red seaweed that belongs to the Bangiaceae family and the *Neoporphyra* or *Pyropia* genera. According to the latest information from Algaebase, the *Neoporphyra* genus comprises 11 species, while the *Pyropia* genus includes 113 species. Notably, some species within these genera such as *Neoporphyra dentata*, *Neoporphyra haitanensis*, *Neoporphyra seriata*, *Neopyropia yezoensis*, *Neopyropia tenera*, and *Pyropia suborbiculata*, are commercially important laver species that are consumed as processed foods [2]. While processed lavers are popular among consumers due to their improved quality, longer shelf life, and convenient preparation, the elimination of morphological features during processing presents a challenge in accurately identifying the species [6]. This may frequently lead to unintentional or intentional substitution and mislabeling of processed laver products. Unintentional substitution may occur due to morphological similarities between closely related species or uncertainties in the common names used for species marketed domestically or internationally [6]. In contrast, intentional substitution is typically performed to increase profitability and may involve the sale of lower-value or endangered species [7]. Therefore, there is a need for an authentication method that can distinguish between different laver species to prevent mislabeling and provide accurate information to consumers.

A dependable and sensitive analytical technique can help to maintain food quality standards and safeguard customers from fraud. Recently, there have been significant advancements in authentication techniques for food fraud, with a focus on DNA- and protein-based technologies [8,9]. DNA-based methods are particularly suitable for analyzing processed foods compared to protein-based methods because DNA is stable at high temperatures and pressures [10]. The most popular DNA-based methods for food authentication include random amplified polymorphic DNA (RAPD), rapid PCR-lateral flow assay, real-time PCR, and multiplex PCR. Each method has its own advantages, disadvantages, and specific applications [11,12,13].

Multiplex PCR can simultaneously detect multiple species within a single reaction tube. A notable advancement in this approach is the integration of capillary electrophoresis with multiplex PCR, which enables the clear differentiation of amplicons with similar sizes, particularly short PCR products. Consequently, this technique has successfully been used to simultaneously detect various target species. However, there is currently no existing multiplex PCR method capable of simultaneously detecting different types of laver species. Therefore, this study aimed to develop a multiplex platform for assessing the vulnerability to adulteration and economic importance of six laver species. Additionally, this study investigated the occurrence of mislabeling and species substitution in commercially available laver products in Asia.

## 2. Materials and Methods

### 2.1. Sample Collection

The six specimens of laver species, including *N. dentata*, *N. haitanensis*, *N. seriata*, *P. suborbiculata*, *N. yezoensis*, and *N. tenera*, were sourced from the National Institute of Biological Resources (NIBR, Incheon, Republic of Korea) and the Ministry of Food and Drug Safety (MFDS, Cheongju, Republic of Korea) (Appendix A). Additionally, 11 other seaweed specimens, including *Pyropia koreana*, *Neopyropia katadae*, *Pyropia kuniedae*, *Pyropia ishigecola*, *Undaria pinnatifida*, *Costaria costata*, *Undaria crenata*, *Saccharina japonica*, *Saccharina sculpera*, *Sargassum fusiforme*, and *Gracilaria vermiculophylla*, were obtained from NIBR. Processed laver products (dried, roasted, seasoned) were purchased from local markets (Gyeonggi and Jeolla Provinces) and online markets. All samples were ground and stored at −20 °C until analysis.

### 2.2. DNA Extraction

To ensure high-quality DNA, six laver specimens and 11 other seaweed specimens underwent a washing step with distilled water to remove oil and salt, followed by grinding in liquid nitrogen using a mortar and pestle. For the commercial method, 25 mg of the ground sample was dissolved in lysis buffer, and genomic DNA extraction was performed using the DNeasy Plant Mini kit (Qiagen, Valencia, CA, USA). The purity and concentration of the extracted DNA were evaluated using a Maestro Nano-spectrophotometer (Maestrogen, Las Vegas, NV, USA).

### 2.3. Primer Design

The species-specific sequences of chloroplast genomes of 16 seaweed species were obtained from GenBank (accessed on June 2023) and gene accession numbers are shown in Table 1. The sequences were aligned using the Clustal Omega program version 1.2.4 with default parameters (gap opening penalty, 6 bits; gap extension, 1 bit). Species-specific primers were designed from ribulose-bisphosphate carboxylase (*rbcL* or *rbcS*) genes using the Primer Design program version 3.0 (Scientific and Educational Software, Durham, NC, USA). Primer specificity was confirmed through in silico analysis using the Basic Local Alignment Search Tool (BLAST) to compare nucleotide sequences within the NCBI database.

### 2.4. PCR Conditions

#### 2.4.1. Singleplex PCR Conditions

For singleplex PCR, a 25 µL reaction mixture comprised 10× buffer, 0.2 mM dNTPs, 0.5 U Hot Start Taq polymerase DNA (Bioneer, Daejeon, Korea), 0.4 µM each primer, and 10 ng of DNA template. Amplification was conducted using a thermal cycler (Astec, Tokyo, Japan) with the following program: preincubation at 94 °C for 5 min; 40 cycles of denaturation at 94 °C for 10 s, annealing at 60 °C for 10 s, and extension at 72 °C for 10 s; and a final extension at 72 °C for 5 min.

#### 2.4.2. Multiplex PCR Conditions

For multiplex PCR, a 25 µL reaction mixture comprised 10× buffer, 0.2 mM dNTPs, 1 U Hot Start Taq polymerase DNA (Bioneer, Daejeon, Republic of Korea), and 10 ng of DNA template. The optimized multiplex PCR was conducted using the six primer sets at the concentrations specified in Table 2. Amplification was performed under the same conditions and instrument as the singleplex PCR.

In this study, six species-specific primer pairs were divided into two sets based on their intended targets. The first set included stone laver species (*N. dentata*, *N. seriata*, and *P. suborbiculata*), while the second set comprised traditional laver species (*N. yezoensis*, *N. tenera*, and *N. haitanensis*). Each primer pair was designed to exhibit an amplicon size difference of at least 40 bp, allowing for the distinction of four PCR amplicons in capillary electrophoresis (set1: *N. dentata*, 274 bp; *N. seriata*, 195 bp; *P. suborbiculata*, 117 bp; 18S rRNA, 89 bp; set2: *N. yezoensis*, 211 bp; *N. tenera*, 169 bp; *N. haitanensis*, 127 bp; 18S rRNA, 89 bp).

#### 2.4.3. Electrophoresis

The amplified product was validated through capillary electrophoresis using an Agilent 2100 Bioanalyzer (Agilent Technologies, Santa Clara, CA, USA) equipped with an Agilent DNA 1000 Kit (Agilent Technologies, Santa Clara, CA, USA). Briefly, each of the 12 wells was loaded with 1 μL of PCR product and 5 μL of size markers. Subsequently, a gel dye mix was applied to the samples in the chip, followed by running the bioanalyzer for analysis.

### 2.5. Specificity and Sensitivity

To assess primer specificity, we used the genomic DNA from six laver species (*N. dentata*, *N. haitanensis*, *N. seriata*, *P. suborbiculata*, *N. yezoensis*, and *N. tenera*) and 11 other seaweed species (*P. koreana*, *N. katadae*, *P. kuniedae*, *U. ishigecola*, *U. pinnatifida*, *C*. *costata*, *U. crenata*, *S*. *japonica*, *S*. *sculpera*, *S*. *fusiforme*, and *G*. *vermiculophylla*). Sensitivity was confirmed through triplicate experiments using serially diluted genomic DNA ranging from 10 ng to 0.01 pg.

### 2.6. DNA Sequencing

PCR product was purified using a QIAquick PCR purification kit (Qiagen, Valencia, CA, USA). Following purification, the product was sequenced using a DNA sequencer (Applied Biosystems, Foster City, CA, USA). To verify the nucleotide sequences, a BLAST search was conducted against the NCBI database.

### 2.7. Statistical Analysis

Each assay was performed three times. The results of replicates are presented as the average value, along with the standard deviation (SD). The electrophoresis results were analyzed using Agilent 2100 Expert software version B.02.11.SI811.

## 3. Results and Discussion

### 3.1. Specificity of the Primers

DNA-based molecular methods are widely employed for the authentication of marine products, including seaweed, due to the stability and recoverability of DNA from extensively processed food [6,15], and successful species identification relies on well-designed primers [16]. Chloroplast DNA, with its multiple copies per cell, is efficient, featuring regions of low intraspecific variation and high interspecific polymorphism, facilitating species identification. Commonly used chloroplast genes for food authentication include *matK*, *rpl16*, *rpoC2*, *trnH-psbA*, *rbcL*, and *rbcS* [17,18]. In this study, we used the *rbcL* and *rbcS* genes, which exhibit an intraspecies conserved and interspecies polymorphic nature. In the dendrogram based on *rbcL* and *rbcS*, six laver species were divided into two groups (Appendix A). Species-specific regions within *rbcL* or *rbcS* gene were selected based on sequence alignment.

The specificity of each primer pair targeting the *rbcL* and *rbcS* genes was confirmed through singleplex PCR with genomic DNA from laver species. Each primer set successfully amplified the targeted sequence of the chloroplast genome to the expected amplicon size. Notably, in singleplex PCR, each primer pair exhibited no cross-reactions with 11 other seaweed species, including *P. koreana*, *N. katadae*, *P. kuniedae*, *U. ishigecola*, *U. pinnatifida*, *C*. *costata*, *U. crenata*, *S*. *japonica*, *S*. *sculpera*, *S*. *fusiforme*, and *G*. *vermiculophylla*. Consequently, each target species-specific primer set exclusively amplified its target species, demonstrating the species specificity of the primers. Additionally, the universal eukaryote-specific primer pair targeting the 18S rRNA gene consistently yielded a PCR product of 89 bp across all species, as expected.

### 3.2. Optimization of Multiplex PCR

To optimize multiplex PCR conditions, various factors, including PCR buffer, dNTPs, Taq DNA polymerase, and primer sets, were adjusted to minimize non-specific interactions [19]. The optimal primer concentrations for each set were determined as follows: for the first set, concentrations of 1.6, 1.8, 0.8, and 0.2 µM were assigned to *N. haitanensis, N*. *tenera, N*. *yezoensis*, and 18S rRNA, respectively; for the second set, concentrations of 0.6, 2.6, 0.32, and 0.4 µM were allocated to *N. dentata, P*. *suborbiculata, N*. *seriata*, and 18S rRNA, respectively. Detailed concentrations of the primers are provided in Table 2. Following this optimization process, the annealing temperature was set to 60 °C, consistent with singleplex PCR. Many previous studies have chosen 40 cycles to detect short amplicons, approximately 250 bp in length, in chloroplast or mitochondrial DNA [20,21,22]. This number of cycles was determined to be optimal, maximizing yield while minimizing non-specific amplification, and is essential for detecting low-abundance targets in processed foods. Increasing the number of cycles may lead to a higher incidence of artificial mutations due to polymerase errors [23]. Conversely, a lower number of PCR cycles may fail to detect trace amounts of the target in processed foods. Therefore, we have decided to use the 40 cycles protocol as outlined in previous research, taking into account the efficiency of amplification, the specificity of the product, and the overall yield.

The specificity of multiplex PCR was validated through capillary electrophoresis for the six laver species in comparison to 11 seaweed species (Figure 1). The use of species-specific primers in multiplex PCR allows for the reliable detection of multiple species [24]. Multiplex PCR emerges as a precise, cost-effective, and sensitive analytical technique, capable of simultaneously identifying numerous species on a single platform. This holds the potential to substantially reduce both analysis cost and time, contributing to the monitoring of species substitution in foods [16]. In this study, multiplex PCR was designed for the routine analysis of species authentication for six lavers. Another method employed herein is capillary electrophoresis, where buffer and high voltage are applied to microfluidic channels with independent electrodes, creating an automated pattern in each well [25]. This method surpasses traditional gel electrophoresis in distinguishing bands of similar sizes, which is particularly advantageous for multiplex PCR [26,27]. Capillary electrophoresis effectively discriminated bands of a similar size, which were simultaneously amplified with four primer sets in a single reaction tube using lab-on-a-chip technology. Under multiplex conditions, each primer set for the six laver species specifically amplified the target species along with the universal eukaryote-specific 18S rRNA gene, showing high resolution between target species. No false-positive amplicons for other species were detected. This validates our multiplex PCR’s sufficient sensitivity and its ability to precisely identify laver species in food. It is expected that the great sensitivity of this assay will make it more effective in accurately distinguishing laver species compared to other methods [28,29].

### 3.3. Sensitivity of Singleplex and Multiplex PCR

The risk of false-positive results due to cross-reactivity or non-specific amplification can be higher in multiplex PCR compared to singleplex PCR assay. Appropriate control and validation should be to address these challenges and ensure the reliability of multiplex PCR results. To ensure both specificity and sensitivity, we initially validated the singleplex PCR and established the multiplex PCR conditions. Subsequently, multiplex PCR exhibited the same level of specificity and sensitivity as the singleplex PCR.

The sensitivity of singleplex and multiplex PCR (set1: *N. dentata*, *N. seriata*, and *P. suborbiculata*; set2: *N. yezoensis*, *N. tenera*, and *N. haitanensis*) was assessed by serially diluting genomic DNA from 10 ng to 0.01 pg. In singleplex PCR, the detection limit for all laver species was 0.1 pg (Figure 2). In multiplex PCR for set1, four peaks were observed in both the gel and electropherogram at a concentration of 0.1 pg. However, at a concentration of 0.01 pg, none of the peaks were detected simultaneously (Figure 3). Similarly, in set2, four peaks were simultaneously detected up to 0.1 pg (Figure 4). Consequently, the sensitivity of multiplex PCR for detecting six laver species was 0.1 pg, equivalent to that of singleplex PCR. This high sensitivity makes it suitable for detecting modified DNA or DNA with low extraction efficacy in processed or field samples.

The sensitivity of the multiplex PCR developed in this study is notably high, comparable to, or exceeding that reported for detection methods in marine products. Notably, there is no existing literature on the sensitivity of *Neoporphyra* and *Pyropia* species detection. For instance, Wilwet et al. (2021) reported PCR methods for authenticating seven shrimp species with a detection limit of 0.1 ng/μL [28]. Moreover, Brenn et al. (2021) reported multiplex real-time PCR methods for detecting four crustaceans [29], achieving a detection limit between 0.2 pg and 2 pg for *Nephrops norvegicus*, *Pleoticus muelleri*, *Litopenaeus vannamei*, and *Penaeus monodon*. The sensitivity of our multiplex PCR method demonstrates significant improvements, affirming its enhanced sensitivity and efficiency over prior approaches. This confirms the adequate sensitivity of our multiplex PCR and its potential in accurately detecting laver species in food. The high sensitivity of this assay is anticipated to facilitate accurate detection and differentiation of laver species.

### 3.4. Laver Authentication Using Multiplex PCR

Laver species, when commercialized as processed products, lose morphological features such as size, shape, or color, making them vulnerable to fraudulent substitutions with lower-value species [11]. To counter such fraud, validated methods are essential for species authentication. Therefore, the multiplex PCR method developed in this study was applicable to authenticate commercial products.

The applicability of our multiplex PCR was assessed using 40 commercially available processed laver products, including 3 dried, 11 roasted, and 26 seasoned items. All samples had undergone heating, pressurization, and freezing, and the seasonings were extensively processed. Among these samples, 9 indicated specific species on their labels, while 31 provided only general commercial names (e.g., laver) without scientific species names (Table 3). For 15% of the products (P26, P30, P31, P32, P34, and P37), the species declared on the labels did not align with those detected by multiplex PCR. The misrecognition of laver species by the manufacturers could be due to their similar morphology, or unintentional mixing might have occurred as a result of cross-contamination. Notably, three products (P31, P32, and P34) labeled as *N. tenera* were found to contain *N. yezoensis*, *N. dentata*, or *P. suborbiculata*. The remaining three products (P26, P30, and P37) labeled as only *N. dentata* were observed to contain a mixture of *N. yezoensis*, *P. suborbiculata*, or *N. seriata*, which were not declared on the label. Given that *N. tenera* and *N. dentata* are two-to-three-times more expensive than other species [30], the inclusion of other laver species in processed products (labeled as *N. tenera* and *N. dentata*) is likely due to economic reasons. Although *N. tenera* has a good taste and high amino acid content, its decreasing distribution and endangered status in both Korea and Japan contribute to its absence from fish farms. Products mislabeled as *N. tenera* should accurately state the species information on their labels. In addition to the PCR product targeting species-specific regions, the universal primer pair for the 18S rRNA gene successfully amplified an 89 bp product for all DNA extracted from the processed laver products. These findings indicate the applicability of our multiplex PCR method as an effective tool for authenticating laver species in processed products.

We developed a rapid and cost-effective multiplex PCR method capable of simultaneously detecting six laver species and confirmed the sequence of PCR products through DNA sequencing. The identity of these products as the target species was then confirmed by conducting a BLAST search against the NCBI nucleotide database, as detailed in Appendix A. All samples that exhibited positive signals in the multiplex PCR were confirmed as the target laver species by BLAST search with >99% identities. The analysis of DNA sequencing for PCR products supports the finding that there was no cross-reaction and misrecognitions with the specific primers.

Recently, Giantsis et al. (2023) presented a multiplex PCR for identifying fish species of the Mullidae family using mitochondrial genes with target lengths ranging from 106 to 438 bp [31]. Similarly, Saetang and Benjakul (2022) proposed a cytochrome b sequence-based PCR for detecting Asian seabass and mangrove red snapper with amplicon lengths between 268 and 480 bp [32]. Wilwet et al. (2021) reported another species-specific PCR with a 204–778 bp target to differentiate seven shrimp species [28]. However, these reports rely on longer amplicons (106–778 bp). Previous research suggests that the stability of PCR-based methods for detecting processed foods depends on the size of the amplicons. Longer target sizes are more susceptible to degradation under conditions encountered during food processing, reducing the efficiency and applicability of the assay in processed products [33]. Addressing this concern, our method kept amplicon lengths within the range of 89 to 274 bp. Uddin et al. (2021) documented that a sheep-specific sequence (236 bp) retained its stability in processed meat samples undergoing boiling (100 °C for 90 min), microwaving (700 W for 30 min), and autoclaving (121 °C for 20 min) [9]. Another study reported that amplicons should be less than 300 bp for the sensitive detection of products severely damaged in high temperature processes [34]. The short amplicons increase specificity as they are less likely to bind to partially complementary sequences in the genome, thereby reducing non-specific binding and amplification. However, they may not capture enough variability to distinguish between similar sequences in highly conserved regions. Nevertheless, we identified regions within short amplicons that possess sufficient variability to distinguish closely related laver species with high specificity. Our approach, using short amplicons, was applied to dried, seasoned, and roasted laver products. Generally, seasoned laver products are produced by heat treatment at 230 °C for 20 min, and then mixed with seasoning. Therefore, our assay was experimentally demonstrated to possess stability and efficiency in products undergoing various processing processes, successfully detecting laver species in processed products.

## 4. Conclusions

In this study, we established an optimized and reliable capillary electrophoresis-based multiplex PCR method for laver authentication. Our method has the advantages of being time-saving and cost-effective as it can simultaneously detect six laver species using six primer sets. The newly developed primers were successfully validated with reference specimens, demonstrating sufficient specificity to distinguish even closely related species from the target species. Furthermore, this assay amplified target species in highly heat-processed products (230 °C for 20 min) using short-length targets. Consequently, it holds the potential to be a practical tool for authenticating laver products, particularly in cases involving degraded specimens. Our method allows for the sensitive and simultaneous detection of multiple target species, making it a suitable assay for broadly screening false labeling of laver products to uncover food adulteration. This will contribute to identifying potential sources of food fraud and, consequently, reduce the risk of encountering fraudulent or adulterated ingredients.

## Figures and Tables

**Figure 1 foods-13-00363-f001:**
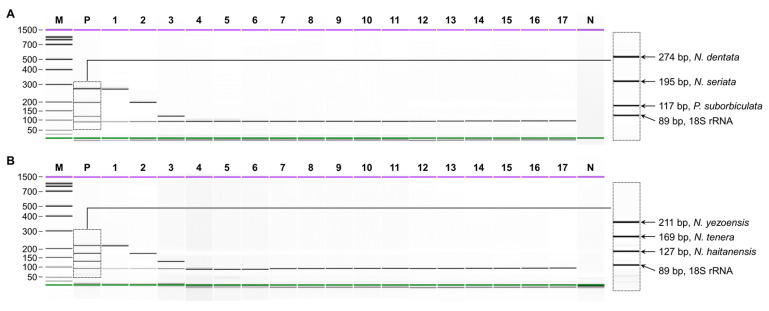
Specificity of multiplex PCR. (**A**) Set1; *N. dentata*, *N. seriata*, *P. suborbiculata*, and 18S rRNA gene. (**B**) Set2; *N. yezoensis*, *N. tenera*, *N. haitanensis*, and 18S rRNA gene. Lane M, 100 bp ladder; lane P, mixture of each target DNA; lanes 1–17, *N. yezoensis*, *N. tenera*, *N. haitanensis*, *N. dentata*, *N. seriata*, *P. suborbiculata*, *P. koreana*, *N. katadae*, *P. kuniedae*, *P. ishigecola*, *U. pinnatifida*, *C. costata*, *U. crenata*, *S. japonica*, *S. sculpera*, *S. fusiforme*, *G. longissima*; lane N, non-template.

**Figure 2 foods-13-00363-f002:**
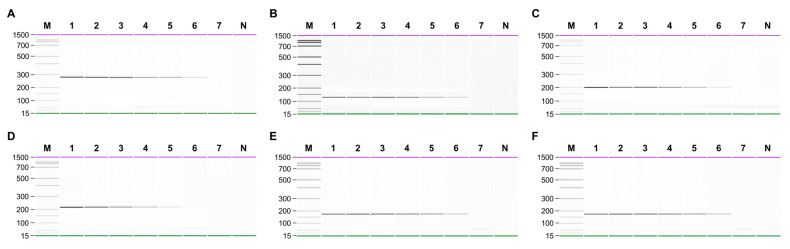
Sensitivity of singleplex PCR for detecting (**A**) *N. dentata*, (**B**) *N. haitanensis*, (**C**) *N. seriata*, (**D**) *N. yezoensis*, (**E**) *N. tenera*, and (**F**) *P. suborbiculata*. Gel-like images of PCR products obtained by serial-diluted genomic DNA of target. Lane M, 100 bp DNA ladder; lanes 1–7, 10, 1, 0,1, 0.01, 0.001, 0.0001, and 0.00001 ng of each target genomic DNA; lane N, non-template.

**Figure 3 foods-13-00363-f003:**
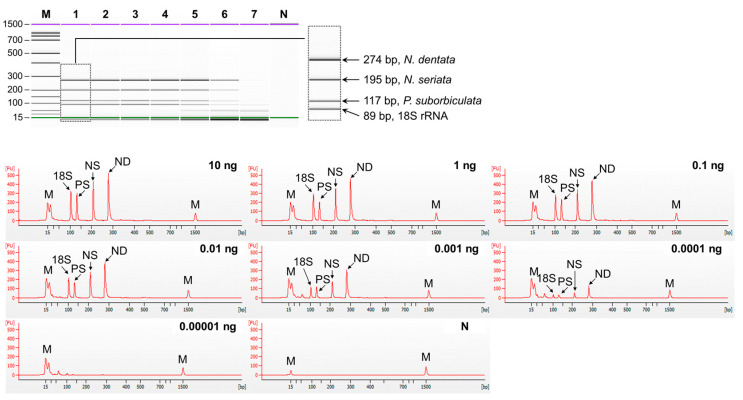
Sensitivity of multiplex PCR of set1 (*N. dentata*, *N. seriata*, *P. suborbiculata*, and 18S rRNA gene). Gel-like (**top**) and electropherogram (**bottom**) images of multiplex PCR products obtained by serial-diluted mixed DNA. Lane M, 100 bp DNA ladder; lanes 1–7, 10, 1, 0,1, 0.01, 0.001, 0.0001, and 0.00001 ng of mixed genomic DNA; lane N, non-template. In the electropherogram, the X-axis represents amplicon size (bp) and the Y-axis represents the measurement response of fluorescence intensity. PS, *P. suborbiculata*; NS, *N. seriata*; ND, *N. dentata*.

**Figure 4 foods-13-00363-f004:**
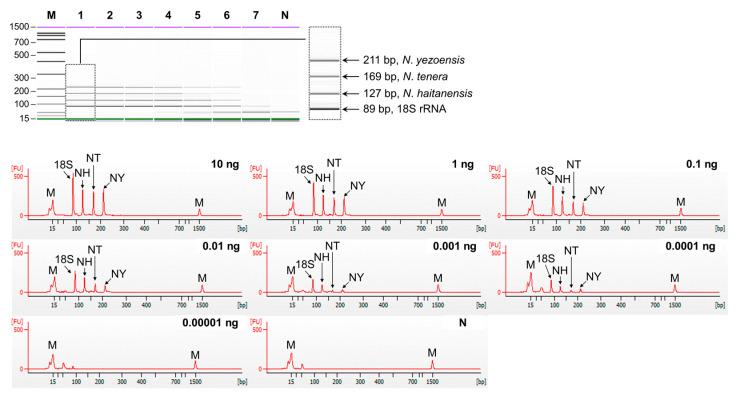
Sensitivity of multiplex PCR of set2 (*N. yezoensis*, *N. tenera*, *N. haitanensis*). Gel-like (**top**) and electropherogram (**bottom**) images of multiplex PCR products obtained by serial-diluted mixed DNA. Lane M, 100 bp DNA ladder; lanes 1–7, 10, 1, 0,1, 0.01, 0.001, 0.0001, and 0.00001 ng of mixed genomic DNA; lane N, non-template. In the electropherogram, the X-axis represents amplicon size (bp) and the Y-axis represents the measurement response of fluorescence intensity. NH, *N. haitanensis*; NT, *N. tenera*; NY, *N. yezoensis*.

**Table 1 foods-13-00363-t001:** Sequence information used for sequence alignment in this study.

Scientific Name	Algae	Accession no. (*rbcL* and *rbcS*)
*Neoporphyra dentata*	Red algae	LC521919.1, LC521919.1
*Neoporphyra haitanensis*	Red algae	KC464603.1, KC464603.1
*Neoporphyra seriata*	Red algae	LC505532.1, LC505532.1
*Pyropia suborbiculata*	Red algae	AB118580.1, AB118580.1
*Neopyropia yezoensis*	Red algae	MT876197.1, MT876197.1
*Neopyropia tenera*	Red algae	AB118576.1, AB118576.1
*Porphyra koreana*	Red algae	LC327005.1, LC327005.1
*Neopyropia katadae*	Red algae	AB118583.1, AB118583.1
*Pyropia kuniedae*	Red algae	LC505521.1, LC505521.1
*Pyropia ishigecola*	Red algae	GQ427224.1
*Undaria pinnatifida*	Brown algae	KP298002.1, KP298002.1
*Costaria costata*	Brown algae	KR336545.1, KR336545.1
*Saccharina japonica*	Brown algae	JQ405663.1, JQ405663.1
*Saccharina sculpera*	Brown algae	JX442492.1, AF318981.1
*Sargassum fusiforme*	Brown algae	MN794016.1, MN794016.1
*Gracilaria vermiculophylla*	Red algae	OP978508.1, OP978508.1

**Table 2 foods-13-00363-t002:** Information of the primers used in this study.

Set	Species	Gene	Primer	Sequence (5′→3′)	Size (bp)	Conc. (µM)	Reference
Set 1	*N. haitanensis*	*rbcS*	HA_F	CCT TCC AGA CCT AAC TGA TGA AC	127	1.6	This study
			HA_R	TCC CCA TAA TTC CCA ATA TGA G			
	*N. tenera*	*rbcL*	TE_F	CTA CTT GAA AGC GAA ACA GAT ATA	169	1.8	This study
			TE_R	CAC CAC CAA ACT GAA GAA CC			
	*N. yezoensis*	*rbcL*	YE_F	GCT GTT AAA GCT CTT CGC TTG	211	0.8	This study
			YE_R	AAT CAA GAC CGC CTT TCA GG			
	18S rRNA	18S rRNA	18S_F	GGT GCA TGG CCG TTC TTA GT	89	0.2	[14]
			18S_R	TGC GCG CAC CTA TTT AGC AG			This study
Set 2	*N. dentata*	*rbcS*	DE_F	GAG CAA ATT AAT AAG CAG CTT ACT TAC	274	0.6	This study
			DE_R	CTG GCT CGT TAG CAG GTC G			
	*P. suborbiculata*	*rbcL*	SU_F	CAG GTG CAA CTG CTA ATA AAG	117	2.6	This study
			SU_R	GTC CAC AAG TTT TAG CTG CA			
	*N. seriata*	*rbcL*	SE_F	CTG GTA AAA ATT ATG GAA GAG TGG TG	195	0.32	This study
			SE_R	TCG CGG CCG TTA CGT TTA AG			
	18S rRNA	18S rRNA	18S_F	GGT GCA TGG CCG TTC TTA GT	89	0.4	[14]
			18S_R	TGC GCG CAC CTA TTT AGC AG			This study

**Table 3 foods-13-00363-t003:** Application of multiplex PCR to processed laver products.

No.	Country (Market)	Type	Labeling	Detection Results by Multiplex PCR ^1^					
				NY	NT	NH	ND	PS	NS
P1	Japan (Online)	Seasoned	Laver	+	−	−	−	−	−
P2	Japan (Online)	Seasoned	Laver	+	−	−	−	−	−
P3	Japan (Online)	Seasoned	Laver	+	−	−	−	−	−
P4	Japan (Online)	Seasoned	Laver	+	−	−	−	−	−
P5	China (Online)	Dried	*N. haitanensis*	−	−	+	−	−	−
P6	China (Online)	Dried	*N. haitanensis*	−	−	+	−	−	−
P7	China (Online)	Dried	*N. haitanensis*	−	−	+	−	−	−
P8	Thailand (Online)	Seasoned	Laver	+	−	−	−	+	−
P9	Thailand (Online)	Seasoned	Laver	+	−	−	−	−	−
P10	Republic of Korea (Gyeonggi)	Seasoned	Laver	+	−	−	+	+	+
P11	Republic of Korea (Gyeonggi)	Seasoned	Laver	+	−	−	+	+	+
P12	Republic of Korea (Gyeonggi)	Seasoned	Laver	+	−	−	+	+	+
P13	Republic of Korea (Gyeonggi)	Seasoned	Laver	+	−	−	+	+	+
P14	Republic of Korea (Gyeonggi)	Seasoned	Laver	+	−	−	−	+	+
P15	Republic of Korea (Gyeonggi)	Seasoned	Laver	+	−	−	−	+	+
P16	Republic of Korea (Gyeonggi)	Seasoned	Laver	+	−	−	−	+	+
P17	Republic of Korea (Gyeonggi)	Seasoned	Laver	+	−	−	−	+	−
P18	Republic of Korea (Gyeonggi)	Seasoned	Laver	+	−	−	−	−	−
P19	Republic of Korea (Gyeonggi)	Seasoned	Laver	+	−	−	−	−	−
P20	Republic of Korea (Gyeonggi)	Seasoned	Laver	+	−	−	−	+	−
P21	Republic of Korea (Gyeonggi)	Roasted	Laver	+	−	−	+	+	−
P22	Republic of Korea (Gyeonggi)	Seasoned	Laver	+	−	−	−	+	−
P23	Republic of Korea (Gyeonggi)	Seasoned	Laver	+	−	−	−	+	−
P24	Republic of Korea (Gyeonggi)	Seasoned	Laver	+	−	−	−	+	−
P25	Republic of Korea (Gyeonggi)	Roasted	Laver	+	−	−	+	+	+
P26	Republic of Korea (Jeolla)	Roasted	*N. dentata*	+	−	−	+	+	−
P27	Republic of Korea (Gyeonggi)	Roasted	Laver	+	−	−	−	+	+
P28	Republic of Korea (Gyeonggi)	Seasoned	Laver	+	−	−	−	−	−
P29	Republic of Korea (Gyeonggi)	Seasoned	Laver	+	−	−	−	+	+
P30	Republic of Korea (Jeolla)	Roasted	*N. dentata*	+	−	−	+	+	−
P31	Republic of Korea (Jeolla)	Roasted	*N. tenera*	+	−	−	−	+	−
P32	Republic of Korea (Jeolla)	Roasted	*N. tenera*	+	−	−	+	+	−
P33	Republic of Korea (Gyeonggi)	Roasted	Laver	+	−	−	+	−	−
P34	Republic of Korea (Jeolla)	Roasted	*N. tenera*	+	−	−	+	+	−
P35	Republic of Korea (Gyeonggi)	Roasted	Laver	+	−	−	−	−	+
P36	Republic of Korea (Gyeonggi)	Seasoned	Laver	+	−	−	−	+	−
P37	Republic of Korea (Jeolla)	Seasoned	*N. dentata*	+	−	−	+	+	+
P38	Republic of Korea (Gyeonggi)	Seasoned	Laver	+	−	−	−	+	−
P39	Republic of Korea (Gyeonggi)	Seasoned	Laver	+	−	−	−	+	−
P40	Republic of Korea (Gyeonggi)	Roasted	Laver	+	−	−	−	−	−

^1^ NY, *N. yezoensis*; NT, *N. tenera*; NH, *N. haitanensis*; ND, *N. dentata*; PS, *P. suborbiculata*; NS, *N. seriata*. Detection results by on-site ultrafast real-time PCR with simple DNA extraction/ultrafast real-time PCR with kit DNA extraction. +, amplified by the corresponding primers; −, not amplified.

## Data Availability

Data is contained within the article or Appendix A.

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
