# Peer review of "Rapid and Simultaneous Authentication of Six Laver Species Using Capillary Electrophoresis-Based Multiplex PCR"

_foods, 2024, doi:10.3390/foods13030363_

Round 1

Reviewer 1 Report

Comments and Suggestions for Authors

kindly modify as suggested

Author Response

Response to Reviewer 1 Comments

Abstract:
Existing: The assay found instances in which less expensive species were mixed in, in addition to identifying the laver species as stated on the commercial label.
Modified as: In addition to detecting the laver species as stated on the commercial label, the assay discovered cases where less expensive species were mixed in.

Response: As you recommended, we revised the sentence in lines 19-21 as follows:

Lines 19-21: In addition to detecting the laver species as stated on the commercial label, the assay discovered cases where less expensive species were mixed in.

Lines 58-59:
Existing: A reliable and sensitive analytical technique can play a crucial role in maintaining food quality standards and protecting consumers against fraud.
Modified as: A dependable and sensitive analytical technique can help to maintain food quality standards and safeguard customers from fraud.

Response: As you recommended, we revised the sentence in lines 57-58 as follows:

Lines 57-58: A dependable and sensitive analytical technique can help to maintain food quality standards and safeguard customers from fraud.

Introduction
Existing: Comprehensive and uptodate

Response: Thank you for your comments.

Sample collection
Existing: Number of specimens collected and what statistical methodology followed while doing collection may be mentioned. How may samples used for DNA isolation etc may be mentioned

Response: As you recommended, we revised or added the sentence in lines 78, 81, 88-89 as follows:

Line 78: The six specimens of laver species

Line 81: Other 11 seaweed specimens

Lines 88-89: To ensure high-quality DNA, six laver specimens and 11 other seaweed specimens underwent a washing step

Statistical analysis
Existing: Information on statistical analysis techniques may be mentioned with soiftweares

Response: As you recommended, we added the sentence in lines 147-150 as follows:

Lines 147-150: 2.7. Statistical analysis

Each assay was performed three times. The results of replicates are presented as the average value, along with the standard deviation (SD). The electrophoresis results were analyzed using Agilent 2100 Expert software version B.02.11.SI811.

Lines 134-136:
Existing: Existing: DNA-based molecular methods are widely employed for the authentication of marine products, including seaweed, due to the stability and recoverability of DNA from extensively processed food [6,14]. Successful species authentication relies on well-designed primers [15].
Modified as: DNA-based molecular methods are widely employed for the authentication of marine products, including seaweed, due to the stability and recoverability of DNA from extensively processed food [6,14] and successful species identification relies on welldesigned primers [15].

Response: As you recommended, we revised the sentence in lines 153-156 as follows:

Lines 153-156: DNA-based molecular methods are widely employed for the authentication of marine products, including seaweed, due to the stability and recoverability of DNA from extensively processed food [6,15] and successful species identification relies on well-designed primers [16].

Lines 155-180:
Existing: More like materials methods and hence only the results part alone may be given and the methodology part may be provided in material and methods
Modified as: This validates our multiplex PCR's sufficient sensitivity and its ability to precisely identify laver species in food. It is expected that this assay's great sensitivity will make it easier to distinguish between different laver species and to detect them accurately

Response: As you recommended, we moved the methodology part to material and methods and revised the sentence.

Lines 120-126: In this study, six species-specific primers were divided into two sets based on their intended targets. The first set included stone laver species (N. dentata, N. seriata, and P. suborbiculata), while the second set comprised traditional laver species (N. yezoensis, N. tenera, and N. haitanensis). Each primer was designed to exhibit an amplicon size difference of at least 40 bp, allowing the distinction of four PCR amplicons in capillary electrophoresis (set1: N. dentata, 274 bp; N. seriata, 195 bp; P. suborbiculata, 117 bp; 18S rRNA, 89 bp; set2: N. yezoensis, 211 bp; N. tenera, 169 bp; N. haitanensis, 127 bp; 18S rRNA, 89 bp).

Lines 209-211: This validates our multiplex PCR's sufficient sensitivity and its ability to precisely identify laver species in food. It is expected that this assay's great sensitivity will make it easier to distinguish between different laver species and to detect them accurately.

Conclusion
Existing: Vague and may be more precise with highlights of the findings and kindly mention the strength and weakness of your present study

Response: As you recommended, we revised the sentence in lines 335-337 and 339-340 as follows:

Lines 335-337: This study has the advantage of time-saving and cost effective, as it can simultaneously detect six laver species using only two multiplex sets.

Lines 339-340: Furthermore, this analysis amplified target species in highly heat-processed products (230°C for 20 min) using short-length targets.

Reviewer 2 Report

Comments and Suggestions for Authors

The manuscript entitled “Rapid and simultaneous authentication of six laver species using capillary electrophoresis-based multiplex PCR” aims to develop a multiplex platform for assessing the vulnerability to adulteration and economic importance of six laver species. Additionally, this study investigates the occurrence of mislabeling and species substitution in commercially available laver products in Asia. I find the manuscript to be a novel contribution to the field, especially in its application for confirming the authenticity of food products using capillary electrophoresis-based multiplex PCR. The method presented holds great promise for practical applications. However, it could be done by the inclusion of a broader range of seaweed species in the analysis. Expanding the scope to include more species could enhance the versatility and applicability of the proposed method, providing a more comprehensive tool for authenticating a wider variety of seaweed products in the future. With the comments:

1- The overall language of the manuscript is commendable; however, I would suggest moderate polishing to enhance clarity and coherence. Some sentences could benefit from smoother transitions or more precise phrasing. Additionally, careful proofreading for grammatical and typographical errors will further improve the overall readability. This adjustment will contribute to a more polished and professional presentation of the valuable research findings.

2- The overall language of the manuscript is commendable; however, I would suggest a moderate polishing to enhance clarity and coherence. Some sentences could benefit from smoother transitions or more precise phrasing. Additionally, careful proofreading for grammatical and typographical errors will further improve the overall readability. This adjustment will contribute to a more polished and professional presentation of the valuable research findings

3- While the selection of processed materials is well-described, it would be beneficial to include local market information to provide context for the readers. This would be important as different regions of the country may use different species as the component of laver. If the geographical distances are diverse show them on a map.

4- A phenotypic picture or a descriptive table of studied seaweed species could be informative.

5- To ensure reproducibility and clarity, please include all pertinent details in the Materials and Methods section. This should encompass specifics such as the Gene ID of utilized sequences, gap penalty and gap extension parameters, and the exact version of the software employed. For online tools, kindly mention the access date. Moreover, provide all information regarding the used reagents (Company, City, and Country of origin).

6- The authors can also provide a dendrogram or PCoA classification of the studied species based on chloroplast sequence barcoding as supplementary material.
Altogether the manuscript is completely eligible for publication in Foods providing addressing above mentioned questions.

Comments on the Quality of English Language

Moderate editing of English language required

Author Response

Response to Reviewer 2 Comments

The manuscript entitled “Rapid and simultaneous authentication of six laver species using capillary electrophoresis-based multiplex PCR” aims to develop a multiplex platform for assessing the vulnerability to adulteration and economic importance of six laver species. Additionally, this study investigates the occurrence of mislabeling and species substitution in commercially available laver products in Asia. I find the manuscript to be a novel contribution to the field, especially in its application for confirming the authenticity of food products using capillary electrophoresis-based multiplex PCR. The method presented holds great promise for practical applications. However, it could be done by the inclusion of a broader range of seaweed species in the analysis. Expanding the scope to include more species could enhance the versatility and applicability of the proposed method, providing a more comprehensive tool for authenticating a wider variety of seaweed products in the future.

With the comments:

1- The overall language of the manuscript is commendable; however, I would suggest moderate polishing to enhance clarity and coherence. Some sentences could benefit from smoother transitions or more precise phrasing. Additionally, careful proofreading for grammatical and typographical errors will further improve the overall readability. This adjustment will contribute to a more polished and professional presentation of the valuable research findings.

Response: As you recommended, we revised the sentence as follows:

Lines 19-21: In addition to detecting the laver species as stated on the commercial label, the assay discovered cases where less expensive species were mixed in.

Lines 57-58: A dependable and sensitive analytical technique can help to maintain food quality standards and safeguard customers from fraud.

Lines 88-89: To ensure high-quality DNA, six laver specimens and 11 other seaweed specimens underwent a washing step

Lines 153-156: DNA-based molecular methods are widely employed for the authentication of marine products, including seaweed, due to the stability and recoverability of DNA from extensively processed food [6,15] and successful species identification relies on well-designed primers [16].

2- The overall language of the manuscript is commendable; however, I would suggest a moderate polishing to enhance clarity and coherence. Some sentences could benefit from smoother transitions or more precise phrasing. Additionally, careful proofreading for grammatical and typographical errors will further improve the overall readability. This adjustment will contribute to a more polished and professional presentation of the valuable research findings

Response: As you recommended, we revised the sentence as follows:

Lines 19-21: In addition to detecting the laver species as stated on the commercial label, the assay discovered cases where less expensive species were mixed in.

Lines 57-58: A dependable and sensitive analytical technique can help to maintain food quality standards and safeguard customers from fraud.

Lines 88-89: To ensure high-quality DNA, six laver specimens and 11 other seaweed specimens underwent a washing step

Lines 153-156: DNA-based molecular methods are widely employed for the authentication of marine products, including seaweed, due to the stability and recoverability of DNA from extensively processed food [6,15] and successful species identification relies on well-designed primers [16].

3- While the selection of processed materials is well-described, it would be beneficial to include local market information to provide context for the readers. This would be important as different regions of the country may use different species as the component of laver. If the geographical distances are diverse show them on a map.

Response: As you recommended, we added local market information in Table 3.

Lines 84-86: Processed laver products (dried, roasted, seasoned) were purchased from local markets (Gyeonggi and Jeolla Provinces) and online markets.

Table 3: We added local market information in Table 3.

4- A phenotypic picture or a descriptive table of studied seaweed species could be informative.

Response: As you recommended, we added the phenotypic picture in Figure S1.

Figure S1: We newly added phenotypic pictures for six laver species in Figure S1.

5- To ensure reproducibility and clarity, please include all pertinent details in the Materials and Methods section. This should encompass specifics such as the Gene ID of utilized sequences, gap penalty and gap extension parameters, and the exact version of the software employed. For online tools, kindly mention the access date. Moreover, provide all information regarding the used reagents (Company, City, and Country of origin).

Response: As you recommended, we revised the sentence in lines 92, 96-99, 116, and 131 as follows:

Line 92: DNeasy Plant Mini kit (Qiagen, Valencia, CA, USA)

Lines 96-99: The species-specific sequences of chloroplast genomes of 16 seaweed species were obtained from GenBank (accessed on June 2023) and gene accession numbers are shown in Table 1. The sequences were aligned using the Clustal Omega program version 1.2.4 with default parameters (gap opening penalty, 6 bits; gap extension, 1 bit).

Line 116: Hot Start Taq polymerase DNA (Bioneer, Daejeon, Korea)

Line 131: Agilent DNA 1000 Kit (Agilent Technologies, Santa Clara, CA, USA)

6- The authors can also provide a dendrogram or PCoA classification of the studied species based on chloroplast sequence barcoding as supplementary material.
Altogether the manuscript is completely eligible for publication in Foods providing addressing above mentioned questions.

Response: As you recommended, we newly analyzed a dendrogram and added the sentence in lines 161-163 and 351-353 as follows:

Lines 161-163: In the dendrogram based on rbcL and rbcS, six laver species were divided into two groups (Figure S2). Species-specific regions within rbcL or rbcS gene were selected based on sequence alignment.

Lines 351-353: Figure S2: The dendrogram based on (A) rbcL and (B) rbcS sequences for 16 seaweed species. The phylogenetic tree was constructed by using the neighbor-joining method within the MEGA software (version 11). The bold text indicates six laver species.

Figure S2: We newly added a dendrogram for rbcL and rbcS sequences for 16 seaweed species in Figure S2.

Reviewer 3 Report

Comments and Suggestions for Authors

This study develops a new capillary electrophoresis-based multiplex PCR technique for identifying different species of seaweed in processed foods, and details its potential applications. Newly designed species-specific primers have been used to detect and obtain specific amplification products for each seaweed species. This theme is of great interest in the field of food science and quality control, especially in the modern context where food adulteration and mislabeling are concerns.

The results and conclusions of the paper are clearly presented. The use of short amplicons has improved the stability of DNA in processed foods and enabled accurate identification of various seaweed species. This approach is expected to contribute to the assurance of food safety and quality.

The results are based on appropriate experimental outcomes and related literature references, demonstrating a scientifically sound and appropriate approach. The detailed explanation of the development and application of the multiplex PCR method is well-founded.

However, based on the content of the paper, I recommend revision, primarily in the following areas:

Commet #1

Impact on specificity and selectivity: More detailed discussion is needed on the impact of using short amplicons on specificity and selectivity. In particular, an explanation of how sufficient selectivity was maintained with the short primer design used in this study would deepen the reader's understanding.

Commet #2

Consideration of misrecognition and cross-reaction: Add a discussion considering other possible misrecognitions and cross-reactions.

Commet #3

Details of PCR experimental conditions: Detailed information is needed on how the optimal PCR conditions were determined. For example, explain why 40 cycles of PCR was chosen as the optimal condition, with reference to experiments or literature that form the basis for this decision.

These revisions would further enhance the superiority and credibility of this report.

Author Response

Response to Reviewer 3 Comments

This study develops a new capillary electrophoresis-based multiplex PCR technique for identifying different species of seaweed in processed foods, and details its potential applications. Newly designed species-specific primers have been used to detect and obtain specific amplification products for each seaweed species. This theme is of great interest in the field of food science and quality control, especially in the modern context where food adulteration and mislabeling are concerns.
The results and conclusions of the paper are clearly presented. The use of short amplicons has improved the stability of DNA in processed foods and enabled accurate identification of various seaweed species. This approach is expected to contribute to the assurance of food safety and quality.
The results are based on appropriate experimental outcomes and related literature references, demonstrating a scientifically sound and appropriate approach. The detailed explanation of the development and application of the multiplex PCR method is well-founded.
However, based on the content of the paper, I recommend revision, primarily in the following areas:

Commet #1

Impact on specificity and selectivity: More detailed discussion is needed on the impact of using short amplicons on specificity and selectivity. In particular, an explanation of how sufficient selectivity was maintained with the short primer design used in this study would deepen the reader's understanding.

Response: As you recommended, we added the sentence in lines 322-332 as follows:

Lines 322-332: The short amplicons increase specificity as they are less likely to bind to partially complementary sequences in the genome, thereby reducing non-specific binding and amplification. However, they may not capture enough variability to distinguish between similar sequences in highly conserved regions. Nevertheless, we identified regions within short amplicons that possess sufficient variability to distinguish closely related laver species with high specificity. Our approach, using short amplicons, was applied to dried, seasoned, and roasted laver products. Generally, seasoned laver product is produced by heat treatment at 230°C for 20 min, and then mixed with seasoning. Therefore, our assay was experimentally demonstrated for its stability and efficiency in products undergoing various processing processes, successfully detecting laver species in processed products.

Commet #2

Consideration of misrecognition and cross-reaction: Add a discussion considering other possible misrecognitions and cross-reactions.

Response: As you recommended, we newly analyzed DNA sequencing to confirm possible misrecognitions and cross-reactions and added the sentence in lines 142-146, 218-223, 276-278, and 300-302 as follows:

Lines 142-146: 2.6. DNA sequencing

PCR product was purified using QIAquick PCR purification kit (Qiagen, Valencia, CA, USA). Following purification, the product was sequenced using a DNA sequencer (Applied Biosystems, Foster City, CA, USA). To verify the nucleotide sequences, a BLAST search was conducted against the NCBI database.

Lines 218-223: The risk of false-positive results due to cross-reactivity or non-specific amplification can be higher in multiplex PCR compared to singleplex PCR assay. Appropriate control and validation should be to address these challenges and ensure the reliability of multiplex PCR results. To ensure both specificity and sensitivity, we initially validated the singleplex PCR and established the multiplex PCR conditions. Subsequently, multiplex PCR exhibits the same level of specificity and sensitivity as the singleplex PCR.

Lines 276-278: The misrecognitions of laver species by the manufacturer could be due to their similar morphology, or unintentional mixing might have occurred as a result of cross-contamination.

Lines 300-302: The analysis of DNA sequencing for PCR products supports the finding that there was no cross-reaction and misrecognitions with the specific primers.

Table S1: We newly added the DNA sequencing data to Table S1.

Commet #3

Details of PCR experimental conditions: Detailed information is needed on how the optimal PCR conditions were determined. For example, explain why 40 cycles of PCR was chosen as the optimal condition, with reference to experiments or literature that form the basis for this decision. These revisions would further enhance the superiority and credibility of this report.

Response: Many previous studies have chosen 40 cycles to detect short amplicons, approximately 250 bp in length, in chloroplast or mitochondrial DNA (Wilai et al., 2020; Travadi et al., 2023; Lee et al., 2021). This number of cycles was determined to be optimal, maximizing yield while minimizing non-specific amplification, and is essential for detecting low-abundance targets in processed foods. Increasing the number of cycles may lead to a higher incidence of artificial mutations due to polymerase errors (Potapov et al., 2017). Conversely, a lower number of PCR cycles may fail to detect trace amounts of the target in processed foods. Therefore, we have decided to use the 40 cycles protocol as outlined in previous research, taking into account the efficiency of amplification, specificity of the product, and overall yield. As you recommended, we added the sentence in lines 183-191 as follows:

Lines 183-191: Many previous studies have chosen 40 cycles to detect short amplicons, approximately 250 bp in length, in chloroplast or mitochondrial DNA [20–22]. This number of cycles was determined to be optimal, maximizing yield while minimizing non-specific amplification, and is essential for detecting low-abundance targets in processed foods. Increasing the number of cycles may lead to a higher incidence of artificial mutations due to polymerase errors [23]. Conversely, a lower number of PCR cycles may fail to detect trace amounts of the target in processed foods. Therefore, we have decided to use the 40 cycles protocol as outlined in previous research, taking into account the efficiency of amplification, specificity of the product, and overall yield.

Reviewer 4 Report

Comments and Suggestions for Authors

The corresponding author Hae-Yeong Kim, has another similar work, titled “On-site ultrafast real-time PCR detection for commercially important seaweeds (lavers) using a microfluidic chip-based system in the farm-to-fork supply chain”, submitted in Food Chemistry, which is also under review. I compared these two manuscript, and I suggest the authors to carefully avoid the repetition part. For instance, both of the two manuscript claimed the novel primer design, which is obviously not the fact. Moreover, in the present work, the reliability of the multiplex assay should also be tested, before validation on commercial products. The superb sensitivity of the multiplex assay should be discussed, and what/why is the so-called significant improvements? It is better to use other method, such as DNA sequencing, to verify the species for commercial products. Long fragment of 274bp is not suitable for processed products.

Comments on the Quality of English Language

The corresponding author Hae-Yeong Kim, has another similar work, titled “On-site ultrafast real-time PCR detection for commercially important seaweeds (lavers) using a microfluidic chip-based system in the farm-to-fork supply chain”, submitted in Food Chemistry, which is also under review. I compared these two manuscript, and I suggest the authors to carefully avoid the repetition part. For instance, both of the two manuscript claimed the novel primer design, which is obviously not the fact. Moreover, in the present work, the reliability of the multiplex assay should also be tested, before validation on commercial products. The superb sensitivity of the multiplex assay should be discussed, and what/why is the so-called significant improvements? It is better to use other method, such as DNA sequencing, to verify the species for commercial products. Long fragment of 274bp is not suitable for processed products.

Author Response

Response to Reviewer 4 Comments

The corresponding author Hae-Yeong Kim, has another similar work, titled “On-site ultrafast real-time PCR detection for commercially important seaweeds (lavers) using a microfluidic chip-based system in the farm-to-fork supply chain”, submitted in Food Chemistry, which is also under review. I compared these two manuscript, and I suggest the authors to carefully avoid the repetition part. For instance, both of the two manuscript claimed the novel primer design, which is obviously not the fact.

Response: The similar manuscript mentioned by the reviewer will no longer be pursued, and this result submitted in Foods is a new primer development. In other papers, the primers used here will be cited.

Moreover, in the present work, the reliability of the multiplex assay should also be tested, before validation on commercial products.

Response: As you recommended, we added the sentence in lines 183-191 as follows:

Lines 183-191: Many previous studies have chosen 40 cycles to detect short amplicons, approximately 250 bp in length, in chloroplast or mitochondrial DNA [20–22]. This number of cycles was determined to be optimal, maximizing yield while minimizing non-specific amplification, and is essential for detecting low-abundance targets in processed foods. Increasing the number of cycles may lead to a higher incidence of artificial mutations due to polymerase errors [23]. Conversely, a lower number of PCR cycles may fail to detect trace amounts of the target in processed foods. Therefore, we have decided to use the 40 cycles protocol as outlined in previous research, taking into account the efficiency of amplification, specificity of the product, and overall yield.

The superb sensitivity of the multiplex assay should be discussed, and what/why is the so-called significant improvements? It is better to use other method, such as DNA sequencing, to verify the species for commercial products.

Response: As you recommended, we newly analyzed DNA sequencing and added the sentence in lines 142-146 and 293-302 as follows:

Lines 142-146: DNA sequencing
PCR product was purified using QIAquick PCR purification kit (Qiagen, Valencia, CA, USA). Following purification, the product was sequenced using a DNA sequencer (Applied Biosystems, Foster City, CA, USA). To verify the nucleotide sequences, a BLAST search was conducted against the NCBI database.

Lines 293-302: DNA sequencing provides detailed genetic information, but is time-consuming and expensive when it comes to detecting specific targets. In contrast, PCR is not only rapid and cost-effective but also highly sensitive, capable of amplifying DNA from a single cell, making it more efficient than DNA sequencing. The PCR products amplified with species-specific primers were sequenced. The identity of these products as the target species was then confirmed by conducting a BLAST search against the NCBI nucleotide database, as detailed in Table S1. All samples that exhibited positive signals in the multiplex PCR were confirmed as the target laver species by BLAST search with > 99% identities. The analysis of DNA sequencing for PCR products supports the finding that there was no cross-reaction and misrecognitions with the specific primers.

Table S1: We newly added the DNA sequencing data to Table S1.

Long fragment of 274bp is not suitable for processed products.

Response: Some previous studies have demonstrated that amplicons of about 270 bp are also suitable for processed products (Uddin et al., 2021; Tafvizi et al., 2016). As you recommended, we added the sentence in lines 318-332 as follows:

Lines 318-332: Uddin et al. (2021) documented that sheep-specific amplicon (236 bp) retained its stability in processed meat samples undergoing boiling (100°C for 90 min), microwaving (700 W for 30 min), and autoclaving (121°C for 20 min) [9]. Another study reported that amplicon should be less than 300 bp for sensitive detection of products severely damaged in the high temperature processes [34].

Round 2

Reviewer 4 Report

Comments and Suggestions for Authors

The modified manuscript can be accepted for publication in Foods.

Author Response

Thank you for your comments.